# Advanced Detection Method for Dengue NS1 Protein Using Ultrasensitive ELISA with Thio-NAD Cycling

**DOI:** 10.3390/v15091894

**Published:** 2023-09-08

**Authors:** Po-Kai Chen, Jyun-Hao Chang, Liang-Yin Ke, Jun-Kai Kao, Chang-Hua Chen, Rei-Cheng Yang, Teruki Yoshimura, Etsuro Ito, Jih-Jin Tsai

**Affiliations:** 1Department of Biology, Waseda University, Tokyo 162-8480, Japan; pokai@akane.waseda.jp (P.-K.C.); jyunhao@suou.waseda.jp (J.-H.C.); 2Department of Medical Laboratory Science and Biotechnology, College of Health Sciences, Kaohsiung Medical University, Kaohsiung 80708, Taiwan; kly@kmu.edu.tw; 3Frontier Molecular Medical Research Center in Children, Changhua Christian Children’s Hospital, Changhua 50006, Taiwan; 96777@cch.org.tw (J.-K.K.); rechya@kmu.edu.tw (R.-C.Y.); 4Department of Post-Baccalaureate Medicine, College of Medicine, National Chung Hsing University, Taichung 402204, Taiwan; changhua@cch.org.tw; 5Changhua Christian Hospital, Changhua 50006, Taiwan; 6Kaohsiung Medical University Chung-Ho Memorial Hospital, Kaohsiung 80756, Taiwan; 7School of Pharmaceutical Sciences, Health Science University of Hokkaido, Hokkaido 061-0293, Japan; yosimura@hoku-iryo-u.ac.jp; 8Graduate Institute of Medicine, Kaohsiung Medical University, Kaohsiung 80708, Taiwan; 9Tropical Medicine Center, Kaohsiung Medical University Hospital, Kaohsiung 80756, Taiwan; 10School of Medicine, College of Medicine, Kaohsiung Medical University, Kaohsiung 80708, Taiwan; 11Division of Infectious Diseases, Department of Internal Medicine, Kaohsiung Medical University Hospital, Kaohsiung 80756, Taiwan

**Keywords:** dengue virus, detection, NS1 protein, nucleic acid application test, ultrasensitive ELISA

## Abstract

Dengue fever, a mosquito-borne disease in tropical and subtropical climates caused by the dengue virus (DENV), has become a major social and economic burden in recent years. However, current primary detection methods are inadequate for early diagnosis of DENV because they are either time-consuming, expensive, or require training. Non-structural protein 1 (NS1) is secreted during DENV infection and is thus considered a suitable biomarker for the development of an early detection method. In the present study, we developed a detection method for the NS1 protein based on a previously reported thio-NAD cycling ELISA (i.e., ultrasensitive ELISA) and successfully achieved a LOD of 1.152 pg/mL. The clinical diagnosis potential of the detection system was also evaluated by using 85 patient specimens, inclusive of 60 DENV-positive and 25 DENV-negative specimens confirmed by the NAAT method. The results revealed 98.3% (59/60) sensitivity and 100% (25/25) specificity, which was in almost perfect agreement with the NAAT data with a kappa coefficient of 0.972. The present study demonstrates the diagnostic potential of using an ultrasensitive ELISA as a low-cost, easy-to-use method for the detection of DENV compared with NAAT and could be of great benefit in low-income countries.

## 1. Introduction

Dengue is a mosquito-borne disease in tropical and subtropical climates caused by the dengue virus (DENV), which belongs to the genus Flavivirus, family *Flaviviridae* [1]. DENV can be grouped into four distinct but closely related serotypes: DENV-1, DENV-2, DENV-3, and DENV-4 [2]. At least 128 countries and approximately 4 billion people around the world are threatened by DENV [3,4]. Epidemic modeling from the World Health Organization estimates 390 million DENV infections per year, 96 million of which manifest clinically [5]. Over the past few years, dengue epidemics have become a critical issue for society and the economy.

Most patients with DENV infection remain asymptomatic, whereas others develop the clinical manifestations of an acute febrile illness ranging from undifferentiated fever to dengue hemorrhagic fever and dengue shock syndrome [6]. Clinically, no specific effective therapy is available for patients infected with DENV, and symptomatic or supportive treatment is generally given [7,8,9]. Hence, DENV infection management strategies are crucial in the acute phase, and early, precise, and rapid diagnosis is necessary for a better prognosis. There are currently four main detection methods against DENV: viral isolation, rapid testing against the DENV protein antigen or IgM and IgG antibodies, serologic testing against IgM and IgG antibodies by enzyme-linked immunosorbent assay (ELISA), and nucleic acid amplification testing (NAAT) against DENV viral RNA [10]. Viral isolation and identification have high specificity but are time-consuming, taking at least 5 days. The rapid test is the fastest and most cost-efficient among the other methods, but the sensitivity and specificity are relatively low [11]. Serologic testing based on IgM and IgG is limited by the number of days of infection, as testing must be delayed until the level of antibodies rises to a detectable level. NAAT has the highest sensitivity and specificity but is expensive, laborious, and prone to false positivity, and it must be conducted by trained personnel [12]. To overcome the disadvantages of these methods, a new detection method that focuses on the DENV non-structural protein 1 (NS1) is required.

NS1 is a highly conserved glycoprotein of all four types of DENV that is expressed on the surface membrane of the infected cells and secretes into the bloodstream [13]. Recent studies demonstrated that the NS1 protein is the main pathogenesis factor of DENV [14]. The serum levels of the secreted NS1 protein correlate significantly with the viral infection severity in clinical patient samples [15,16]. Furthermore, the elevated plasma NS1 protein concentrations might indicate more rapid disease progression to dengue hemorrhagic fever compared with mild dengue fever in patients with lower NS1 levels [16]. The NS1 protein levels in the bloodstream of patients with dengue hemorrhagic fever/dengue shock syndrome are as high as 50 µg/mL [16]. NS1 protein secretion during DENV infection is established, making the NS1 protein a promising and suitable biomarker for the development of an early detection method.

In the present study, we developed an ultrasensitive ELISA coupled with the thionicotinamide-adenine dinucleotide (thio-NAD) cycling as the signal amplification system, which has already been applied for the detection of several different pathogens, to detect the NS1 protein [17,18,19]. The results obtained by our method were compared with the clinical diagnostic values obtained by NAAT using clinical patient specimens.

## 2. Materials and Methods

### 2.1. Patient Specimens

We obtained permission to use human specimens from the Kaohsiung Medical University Chung-Ho Memorial Hospital (#KMUHIRB-E(П)-20180232) and Waseda University (#2019-075 and #2022-BS001). A total of 85 sera specimens were collected by the Kaohsiung Medical University Hospital, including 60 patient sera specimens that contained all four types of DENV, and 25 dengue-negative sera specimens checked by NAAT were obtained from the Kaohsiung Medical University Chung-Ho Memorial Hospital. See Appendix A: Patient specimens overview in Appendix A for the detailed information of specimens.

### 2.2. Reagents and Chemicals

The recombinant DENV type 2 NS1 protein (Ab181966, produced in a human cell line) was purchased from Abcam (Boston, MA, USA). The antigen was stored at −80 °C as a 10 ng/mL solution in a 1× tris-buffered saline (TBS). The primary (capture) antibody (BMRdn008) and the secondary (detection) antibody (BMRdn012) for the DENV NS1 protein were gifts from Bio Matrix Research (Chiba, Japan). The alkaline phosphatase (ALP) labeling kit (LK13) that used for secondary antibody ALP labeling was purchased from Dojindo Laboratories (Kumamoto, Japan). For the cycling reagents, NADH was purchased from Sigma–Aldrich (N1161-10VL; St. Louis, MO, USA), and 3α-hydroxysteroid dehydrogenase (3α-HSD) was purchased from Asahi Kasei Pharma (T-58; Tokyo, Japan). Thio-NAD was purchased from Oriental Yeast (44104001; Tokyo, Japan). A substrate, 17β-methoxy-5β-androstan-3α-ol 3-phosphate, was synthesized by Teruki Yoshimura.

### 2.3. Ultrasensitive ELISA with Thio-NAD Cycling

The ultrasensitive ELISA was first designed in 2014 [17], and this procedure was applied with slight modifications in the present study. A sandwich ELISA using primary and secondary antibodies to detect antigens was coupled with a thio-NAD cycling method (Figure 1). Two enzymes, ALP and 3α-HSD, constitute the signal amplification system. The enzyme cycling reaction conducted by 3α-HSD includes a forward reaction and a reverse reaction. In the forward reaction, 3α-hydroxysteroid, an androsterone derivative, is produced by ALP labeled on the secondary antibody through the hydrolysis of 3α-hydroxysteroid 3-phosphate. Subsequently, 3α-hydroxysteroid is oxidized to 3-ketosteroid by 3α-HSD using thio-NAD as a cofactor. In the reverse reaction, 3-ketosteroid is reduced to 3α-hydroxysteroid by 3α-HSD using NADH as a cofactor. During the enzyme cycling reaction, thio-NAD was transferred to thio-NADH and accumulated continuously, while thio-NADH can be measured optically in real-time at 405 nm by a simple microplate reader. In both conventional ELISA and enzymatic cycling, the signal is a linear function of the measurement time. However, when these two methods are combined, the signal resulting from the absorbance measurement of thio-NADH molecules increases triangularly, i.e., 1, 1 + 2, (1 + 2) + 3, (1 + 2 + 3) + 4..., and thus can rapidly amplify the signal substance in a short time and reach the standard of ultrasensitive in zeptomole level [20].

A 100 μL solution of primary antibody adjusted to 4 μg/mL in 50 mM Na_2_CO_3_ (pH 9.6) was added to each well of the microplates and incubated for 1 h at room temperature and then washed three times. The wash buffer contained 0.05% Tween 20 in TBS. Next, each well was incubated with a 300 μL blocking solution of 1% bovine serum albumin (BSA) in TBS for 1 h at room temperature and then washed three times. After blocking, 100 μL of the recombinant DENV NS1 protein in TBS containing 0.1% BSA was added to each well to achieve gradient concentrations of 200 pg/mL, 100 pg/mL, 50 pg/mL, 25 pg/mL, and 12.5 pg/mL; 0.1% BSA/TBS was used as a blank. The solutions were incubated for 1 h at room temperature or overnight at 4 °C and then washed nine times. Patient specimens were diluted 1000 times by 0.1% BSA/TBS and added instead of the recombinant protein (see Section 2.5). The secondary antibody conjugated with ALP was diluted by TBS containing 0.05% Tween 20 and 0.1% BSA, added to each well with 100 μL, and then washed nine times. Finally, 100 μL of the thio-NAD cycling solution containing 10 U/mL 3α-HSD, 0.4 mM 17β-methoxy-5β-androstan-3α-ol 3-phosphate, 1.0 mM NADH, and 2.0 mM thio-NAD in 100 mM Tris-HCL was added into each well and measured with a microplate reader (Corona Electric SH-1000; Ibaraki, Japan). The accumulated thio-NADH was measured at 405 nm, and absorbance at 660 nm was used as a reference for background correction.

### 2.4. Spike-and-Recovery Test

For accurate detection of DENV in patient sera specimens, a spike-and-recovery test was performed to examine whether the ultrasensitive ELISA method was affected by the specimen’s matrix, such as proteins in sera. To conduct the spike-and-recovery test, a final concentration of 50 pg/mL of the recombinant DENV NS1 protein was added to diluted solutions of standard control sera (06903; Shimadzu Diagnostics, Tokyo, Japan). The spike and recovery ratio (80–120%) was then evaluated to determine the suitable dilution ratio for patient specimens.

### 2.5. Statistical Analysis

The experimental data were obtained by subtracting the mean absorbance value of the blank (i.e., 0.1% BSA/TBS only) according to the corresponding time point and concentration. The limit of detection (LOD) was derived from the mean value of the blanks plus the 3-fold standard deviation (SD) of the blanks. The limit of quantification (LOQ) was derived from the same mean value of the blanks plus the 10-fold SD of the blanks. Regarding patient specimen analysis, the signal-to-cutoff (S/CO) ratio served as the standard of the infection status, whereas the cutoff value was the mean value of the blanks plus 3 SD, S/CO ≥ 1 was considered dengue-positive and S/CO < 1 was considered dengue-negative. The Kappa statistic was performed on SPSS statistics (SPSS, Chicago, IL, USA) to determine consistency among the results from the ultrasensitive ELISA and NAAT.

## 3. Results

### 3.1. LOD and LOQ of the Ultrasensitive ELISA against the Recombinant DENV Type 2 NS1 Protein

The ultrasensitive ELISA was performed using the recombinant DENV type 2 NS1 protein as the target protein (Figure 2). A total of 60 min was required to complete the enzyme cycling and ELISA reader measurement, and the time point of 55 min was selected. After analysis, a linear calibration curve was obtained and expressed as *y* = 1.23 × 10^−3^ *x*, *R*^2^ = 0.998. The LOD of the NS1 protein obtained statistically by calculating the 3 SD of the blank was 1.152 pg/mL, and the LOQ as 10 SD of the blank was 3.841 pg/mL. Because the assay volume was 100 μL/well and the molecular mass of the antigen was 40 kDa, the LOD could be calculated into 2.88 × 10^−18^ moles/assay, which met the standard of ultrasensitivity, while the LOQ was 9.60 × 10^−18^ moles/assay.

### 3.2. Spike-and-Recovery Test

Before further evaluation of the clinical diagnostic value of the ultrasensitive ELISA method, the dilution ratio of the patient sera specimen was determined by conducting the spike-and-recovery test. The dilution factor was 1:10, 1:100, or 1:1000, whereas the spike was 50 pg/mL of the NS1 protein (Table 1). Although a dilution factor of 1:10 showed the most suitable recovery ratio of 110%, the number of patient specimens provided by the Kaohsiung Medical University Hospital was not sufficient. Thus, we decided to use a dilution factor of 1:1000, which showed a recovery ratio of 108%.

### 3.3. Comparison between NAAT and the Ultrasensitive ELISA for Patient Specimens

As shown in Table 1, 60 specimens were dengue-positive, and 25 were dengue-negative in NAAT. The NAAT cycling threshold (CT) value of those dengue-positive specimens ranged from 12.42 to 31.41. In the ultrasensitive ELISA, 59 specimens were correspondingly positive to the NAAT results, whereas 25 specimens were completely correspondingly negative to the NAAT results. Compared with NAAT, the sensitivity and specificity of the ultrasensitive ELISA were 98.3% and 100%, respectively (Table 2). Of 60 NAAT-confirmed dengue-positive patient specimens, only 1 specimen was negative in the ultrasensitive ELISA. The NAAT data showed that the specimen was a type 4 DENV infection case with a CT value of 21.59. The results of the ultrasensitive ELISA were in almost perfect agreement with the NAAT results, with a kappa value of 0.972 (95% CI: 0.917–1.0).

## 4. Discussion

We successfully established a new ultrasensitive detection method against all types of DENV viruses based on ELISA. The ultrasensitive ELISA coupled with the thio-NAD cycling method obtained an outstanding LOD against the DENV NS1 protein. The clinical application of the method was also assessed using multiple patient specimens, revealing a 98.3% sensitivity and 100% specificity against all four types of DENV. Notably, our ultrasensitive ELISA method detection requires only 1 μL of the patient specimen.

NS1 is a multifaceted enigmatic viral protein in the *Flaviviridae* family of viruses, and its structure contains multiple similar parts across different flaviviruses, which might induce the cross-reactivity of antibodies [21]. Regarding antibodies used in the present study, the manufacturer (Bio Matrix Research) conducted a cross-reactivity test and described it as “almost no cross-reaction to JEV, West Nile virus, Yellow fever virus, and Zika virus NS1 recombinant proteins” [22].

Previous studies showed that the DENV NS1 antigen in blood circulation can be detected in the sera of infected individuals by the first day after the onset of fever (i.e., acute phase), as early as viral RNA [16,23,24]. Thus, DENV NS1 is a suitable target for early diagnosis of DENV infection, and this method is easier and faster than NAAT, which is expensive, requires training, and will be particularly useful in low-income countries. For these reasons, our ultrasensitive ELISA against DENV NS1 was designed to realize an easier early diagnosis of DENV infection.

There are several commercialized DENV NS1 ELISA kits, e.g., the Platelia Dengue NS1 Ag kit (Bio-Rad, Hercules, CA, USA) and the Dengue Early ELISA kit (Panbio, Brisbane, Australia). Previous studies showed that the sensitivity of the Platelia Dengue NS1 Ag kit ranges from 63 to 94%, with specificity ranging from 98.4 to 100% [23,25,26,27]. The sensitivity of the Panbio Dengue Early ELISA kit ranges from 65.5 to 96.0% depending on the type of DENV, and the specificity is 100% [28]. In the present study, our ultrasensitive ELISA successfully detected all four types of DENV with a 98.3% (59/60) sensitivity and 100% (25/25) specificity (25/25). Moreover, our ultrasensitive ELISA reached an LOD of 1.152 pg/mL, which is better than the commercial Platelia dengue NS1 ELISA, which has a LOD of 977 pg/mL [29]. 

Regarding NAAT, it has remained the gold standard for DENV diagnosis to date because of its superior sensitivity and specificity [30]. The Center for Disease Control and Prevention in the USA provides the CDC DENV-1-4 Real-time RT-PCR Multiplex Assay for detection and serotype identification [31]. Upon receiving the patient specimen, they recommend extracting the RNA using a QIAamp DSP Viral RNA Mini kit (Qiagen, Hilden, Germany), which is not only costly and time-consuming but is also limited by the volume of the patient specimen. In addition, the assay results depend on the amount and quality of template RNA purified from human specimens [32], indicating that the result is highly dependent on the operator, who must be trained. Compared with the NAAT method, our ultrasensitive ELISA method is easy to perform and requires no further sample preparation, and the detection can be started immediately after receiving a small sample. These advantages indicate that our ultrasensitive ELISA method could potentially meet the need for a simpler and less expensive detection method than NAAT.

Aside from the ELISA and NAAT, an advanced method was developed in 2018 against the DENV RNA sequence by tandem toehold-mediated displacement reactions (tTMDR) [33]. This detection method is also focused on amplifying the signal and could detect DENV with a limit of as low as six copies of RNA per sample. However, because the target antigen of this method is different from our study, we could not compare the detection limit with this method.

## 5. Conclusions

This ultrasensitive ELISA method is rapid, easy to perform, and requires no professional trainees to operate, indicating that the newly established DENV detection method has the potential to replace the current major DENV detection methods and the ability to solve the need for a low-cost, easy-to-use DENV detection method compared with the NAAT, especially for use in low-income countries.

## Figures and Tables

**Figure 1 viruses-15-01894-f001:**
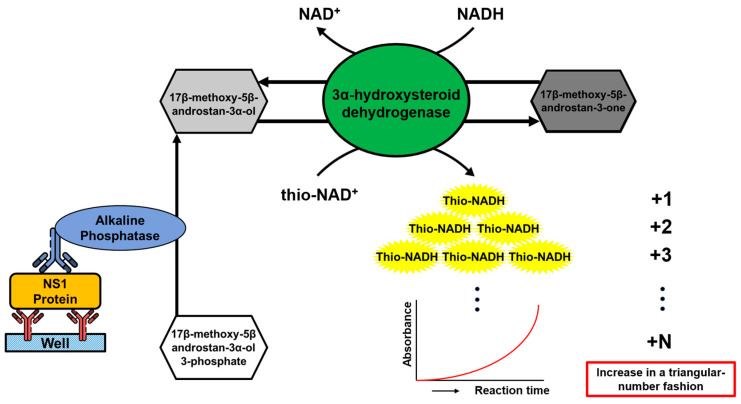
Illustration of ultrasensitive ELISA with thio-NAD cycling. A pair of antibodies was used for capturing the NS1 protein in the sandwich ELISA, and alkaline phosphatase was labeled on the secondary antibody. Aside from the antibodies, an androsterone derivative, 3α-hydroxysteroid dehydrogenase, thio-NAD, and NADH were used to construct the thio-NAD enzyme cycling system. During the thio-NAD cycling reaction, thio-NADH constantly accumulated in a triangular number fashion and could be directly measured at an absorbance of 405 nm.

**Figure 2 viruses-15-01894-f002:**
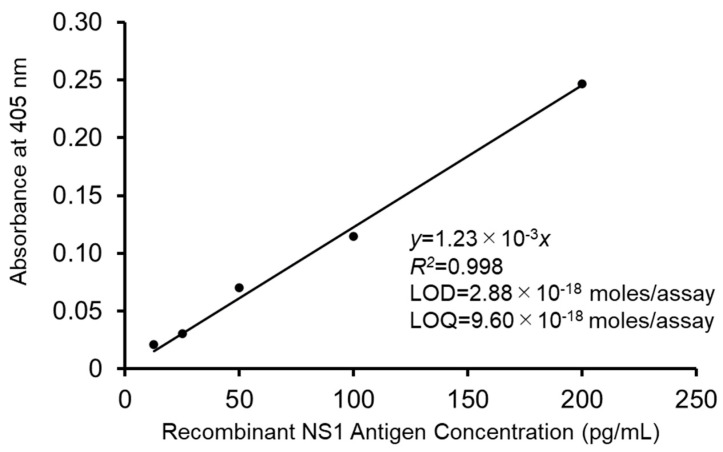
Linear calibration curve obtained from the absorbance for the recombinant DENV type 2 NS1 protein using the thio-NAD cycling ELISA. The absorbance was obtained from the cycling reaction time designated in the text. The experiment was performed with four sets of wells. The antigen was applied in the range of 12.5–200 pg/mL. See Appendix A: Raw data of Figure 2 in Appendix A for more detailed information.

**Table 1 viruses-15-01894-t001:** Spike-and-recovery test result with standard control serum and NS1 protein.

Dilution Factor	Observed (pg/mL) × Dilution Factor	Expected (pg/mL) (Neat Value)	Recovery%
Neat	50.0	50.0	100
1:10	59.7		110
1:100	43.7		80
1:1000	59.0		108

**Table 2 viruses-15-01894-t002:** Comparison of detection results between ultrasensitive ELISA and NAAT.

Ultrasensitive ELISA	NAAT		Total
	Positive	Negative	
Positive	59	0	59
Negative	1	25	26
Total	60	25	85

## Data Availability

The datasets generated and/or analyzed in the present study are available from the corresponding authors upon reasonable request.

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
