# Peer review of "Advanced Detection Method for Dengue NS1 Protein Using Ultrasensitive ELISA with Thio-NAD Cycling"

_viruses, 2023, doi:10.3390/v15091894_

Round 1

Reviewer 1 Report

The authors have realized cycling enhancement in ELISA based on hydrosteroid dehydrogenase use and demonstrated very low detection limit for the case of dengue NS1 antigen detection. The work is in the thematic field of the Viruses journal and may be published in it. However, several issues (especially items 3-5, 9, 10 below) need clarification:

1. The Abstract now is mainly focused on overall characterization of dengue fever and its diagnostics with specification of issues that are over the manuscript. The proposed principle of enhancement is ELISA is not specified, evaluated, and moreover, it not ever named in the Abstract. The authors should reduce statements of common nature and present more focused description of the proposed changes in ELISA protocol. Please indicate also quantity of the tested specimens; the given diagnostic sensitivity and specificity reflect the work with concrete group of patients rather than a standalone property of the method.

2. The Introduction should specify better the existing ELISAs for the detection of specific antigens of the dengue virus and limitations of these ELISAs. Actually the authors indicate that level of the NS1 protein in bloodstream can reach 50 mkg/mL, but do not specify appropriate cut-off level to detect practically all cases of the disease. So the necessity to reach such low LOD as 1.152 pg/mL is not grounded.

3. The ELISA protocol (Section 2.3) contains indication that biosamples should be diluted 1,000 times. The reasons for this decision are not clear. This value is much higher as compared with the common practice of ELISA. Not that the proposed dilution eliminates advantages of high-sensitive detection for NS1 protein in pure solutions, as well as the minimal content of the antigen that can be detected in biosamples occurs to be 1,000 times higher. Taking into account this factor the given comparison with the commercial Platelia ELISA (lines 236-238) is not correct.

4. The authors have tested experimentally solutions of the NS1 protein with concentrations starting from 12.5 pg/mL. So the statement about the possibility to detect up to 1.152 pg/mL (an order of magnitude lower concentration) is based only on theoretical calculations. To ground better this conclusion, it will be reasonable to test additionally preparation(s) with some concentration(s) of the NS1 protein between 1.152 and 12.5 pg/mL.

5. The main idea differing the authors' development from earlier works for dengue diagnostics, i.e. the use of thio-NAD cycling, is hidden inside the EISA protocol and almost is not commented in the manuscript. The mechanism of this enhancement is not specified, and gain in sensitivity over traditional no enhanced ELISA remains unclear. So the text should contain additional consideration of this issue and comparison with the common ELISA protocol.

6. Nine-fold washing (line 120) is not typical for ELISA practice. Usual protocols are limited by 3-4 washing cycles between the stages and by this way are somewhat more time- and reactants-consuming. What were the reasons of the authors to use such protocol?

7. The authors have stated their choice of duration for ELISA measurements (lines 163-164), but the reasons for this decision are not grounded without presentation of experimental data comparing different durations.

8. Fig. 2 and Table 1 should contain error bars for experimental points based on repeated measurements.

9. The used equations of common practice for LOD and LOQ (lines 166-167) state their difference in 3.3 times, whereas the calculated values (line 167, lines 169-170) differ in almost 7 times. This strange situation with the key parameters of the assay needs careful revision.

10. Fig. 3 should be accomplished by raw data of ELISA measurements (for example, as a supplementary table) to be sure in reliability of the discussed effects. What were the concentrations of the viruses used for the comparison?  The commonly used calculation of cross-reactivity values (see https://link.springer.com/article/10.1007/s00216-023-04846-w as an example) will be more informative parameter.

Author Response

Comment 1. The Abstract now is mainly focused on overall characterization of dengue fever and its diagnostics with specification of issues that are over the manuscript. The proposed principle of enhancement is ELISA is not specified, evaluated, and moreover, it not ever named in the Abstract. The authors should reduce statements of common nature and present more focused description of the proposed changes in ELISA protocol. Please indicate also quantity of the tested specimens; the given diagnostic sensitivity and specificity reflect the work with concrete group of patients rather than a standalone property of the method.

Reply 1. Thank you for your comments. We changed the abstract to the following:

Dengue fever, a mosquito-borne disease in tropical and subtropical climates caused by the dengue virus (DENV), has become a major social and economic burden in recent years. However, current primary detection methods are inadequate for early diagnosis of DENV because they are either time-consuming, expensive, or require training. Nonstructural protein 1 (NS1) is secreted during DENV infection and is thus considered a suitable biomarker for the development of an early detection method. In the present study, we developed a detection method of NS1 protein based on a previously reported thio-NAD cycling ELISA (i.e., ultrasensitive ELISA), and successfully achieved an LOD of 1.152 pg/mL. The clinical diagnosis potential of the detection system was also evaluated by using 85 patient specimens, inclusive of 60 DENV-positive and 25 DENV-negative specimens confirmed by the NAAT method. The results revealed 98.3 % (59/60) sensitivity and 100 % (25/25) specificity, which was in almost perfect agreement with the NAAT data with a kappa coefficient of 0.972. The present study demonstrates the diagnostic potential of using an ultrasensitive ELISA as a low-cost, easy-to-use method for the detection of DENV compared with NAAT, and could be of great benefit in low-income countries.

Comment 2. The Introduction should specify better the existing ELISAs for the detection of specific antigens of the dengue virus and limitations of these ELISAs. Actually the authors indicate that level of the NS1 protein in bloodstream can reach 50 mkg/mL, but do not specify appropriate cut-off level to detect practically all cases of the disease. So the necessity to reach such low LOD as 1.152 pg/mL is not grounded.

Reply 2. In Introduction, we have focused on all kinds of detection methods against dengue virus, not just ELISA. Hence, in Discussion, we compared our ultrasensitive ELISA with the 2 existing commercial ELISA kits for dengue NS1 detection.

Comment 3. The ELISA protocol (Section 2.3) contains indication that biosamples should be diluted 1,000 times. The reasons for this decision are not clear. This value is much higher as compared with the common practice of ELISA. Not that the proposed dilution eliminates advantages of high-sensitive detection for NS1 protein in pure solutions, as well as the minimal content of the antigen that can be detected in biosamples occurs to be 1,000 times higher. Taking into account this factor the given comparison with the commercial Platelia ELISA (lines 236-238) is not correct.

Reply 3. The dilution ratio was decided by the spike-and-recovery test, which was shown in Section 3.3 as “Although a dilution factor of 1:10 showed the most suitable recovery ratio of 110 %, the amounts of the patient specimens provided by Kaohsiung Medical University Hospital were not sufficient. Thus, we decided to use a dilution factor of 1:1000, which showed a recovery ratio of 108 %.” As you know well, the results of spike-recovery test should be within 80-120 %. We could detect the NS1 in 1000 times dilution and get a LOD of 1.152 pg/mL, whereas Platelia dengue NS1 ELISA could only obtain a LOD of 977 pg/mL under 4 times dilution (according to the protocol provided), the comparison is clear in our opinion.

Comment 4. The authors have tested experimentally solutions of the NS1 protein with concentrations starting from 12.5 pg/mL. So, the statement about the possibility to detect up to 1.152 pg/mL (an order of magnitude lower concentration) is based only on theoretical calculations. To ground better this conclusion, it will be reasonable to test additionally preparation(s) with some concentration(s) of the NS1 protein between 1.152 and 12.5 pg/mL.

Reply 4. Statistical estimation for LOD is sufficient to prove the validity for the methods. Please kindly refer to Section 2.6 Statistical Analysis.

Comment 5. The main idea differing the authors' development from earlier works for dengue diagnostics, i.e. the use of thio-NAD cycling, is hidden inside the EISA protocol and almost is not commented in the manuscript. The mechanism of this enhancement is not specified, and gain in sensitivity over traditional no enhanced ELISA remains unclear. So the text should contain additional consideration of this issue and comparison with the common ELISA protocol.

Reply 5. We used Figure 1 to roughly explain the idea of thio-NAD cycling’s mechanism, but we now added more description as follows:

Two enzymes, ALP and 3α-HSD, constitute the signal amplification system. The enzyme cycling reaction conducted by 3α-HSD includes a forward reaction and a reverse reaction. In the forward reaction, 3α-hydroxysteroid, an androsterone derivative, is produced by ALP labeled on the secondary antibody through the hydrolysis of 3α-hydroxysteroid 3-phosphate. Subsequently, 3α-hydroxysteroid is oxidized to 3-ketosteroid by 3α-HSD using thio-NAD as a cofactor. In the reverse reaction, 3-ketosteroid is reduced to 3α-hydroxysteroid by 3α-HSD using NADH as a cofactor. During the enzyme cycling reaction, thio-NAD was transferred to thio-NADH and accumulated continuously, while thio-NADH can be measured optically in real time at 405 nm by a simple microplate reader. In both conventional ELISA and enzymatic cycling, the signal is a linear function of the measurement time. However, when these two methods are combined, the signal resulting from the absorbance measurement of thio-NADH molecules increases triangularly, i.e., 1, 1+2, (1+2)+3, (1+2+3)+4..., and thus can rapidly amplify the signal substance in a short time and reach the standard of ultrasensitive in zeptomole level.

Comment 6. Nine-fold washing (line 120) is not typical for ELISA practice. Usual protocols are limited by 3-4 washing cycles between the stages and by this way are somewhat more time- and reactants-consuming. What were the reasons of the authors to use such protocol?

Reply 6. Surely, we can use the protocol by 3-4 washing cycles, however, after multiple attempts, we noticed that washing 9 times after antigen binding and secondary antibody binding could get us a more stable absorbance value for each well, and that is why we chose to use 9 times of washing.

Comment 7. The authors have stated their choice of duration for ELISA measurements (lines 163-164), but the reasons for this decision are not grounded without presentation of experimental data comparing different durations.

Reply 7. The reason we chose the 55 min time point is that at 60 min the signal for the highest concentration of NS1 antigen (200 pg/mL) would have begun to saturate, which would affect the analysis. We added the raw data in Supplementary Materials.

Comment 8. Fig. 2 and Table 1 should contain error bars for experimental points based on repeated measurements.

Reply 8. If we add the error bars to Figure 2, it will contain too much information. We thus showed the raw data in Supplementary Materials.

Comment 9. The used equations of common practice for LOD and LOQ (lines 166-167) state their difference in 3.3 times, whereas the calculated values (line 167, lines 169-170) differ in almost 7 times. This strange situation with the key parameters of the assay needs careful revision.

Reply 9. Thank you for pointing out this crucial problem. We made a mistake on the LOQ calculation and revised it in both Figure 2 and the text.

Comment 10. Fig. 3 should be accomplished by raw data of ELISA measurements (for example, as a supplementary table) to be sure in reliability of the discussed effects. What were the concentrations of the viruses used for the comparison?  The commonly used calculation of cross-reactivity values

(see https://link.springer.com/article/10.1007/s00216-023-04846-w as an example) will be more informative parameter.

Reply 10.

The results of cross-reactivity tests were provided by Bio Matrix Research, which offered the antibodies, and were not calculated by us. So they cannot be expressed as Reviewer 1 suggested. Therefore, Figure 3 was removed from the revised version. Instead, we decided to discuss this point a bit in Discussion. See page 6.

Thank you for your comments.

Reviewer 2 Report

The manuscript titled “Advanced detection method for dengue NS1 protein using ultrasensitive ELISA with Thio-NAD cycling” used a reported signal amplification method to detect dengue NS1 protein. The manuscript provided a useful tool for rapid dengue detection. However, there are several drawbacks that need to be remedied before being published. Thus, I recommend the major revisions outlined below.

1.       In Figure 1, the enzyme-catalyzed thio-NAD cycling should include more time points. The authors just showed the initial time points for the reaction. Longer time period to show the maximum signal should be included. 2.      This manuscript tried to develop a new tool for NS1 protein detection. Thus, the original detection results to show the signal density should be included in Figures 2 and 3.   Minor changes:   3.      Line 58, “testing against IgM and IgG antibodies” was mentioned twice. But the antigen test was missed. 4.      Lines 61 – 63, “ The rapid test ….are relatively low” needs references. 5.      Line 121, 3a-HSD needs the full name. 6.      In the discussion part, more new technologies should be discussed. Recently, tandem toehold-mediated displacement reactions (tTMDR) were developed to detect four serotypes of dengue virus with a limit of as low as six copies of RNA per sample (Chem. Commun. 2018, 54, 968–971; ACS Infect. Dis. 2019, 5, 1907–1914).

7.      Reference 29 was missed.

The manuscript was well written. Some sentences need rewriting to improve the quality of the English language, such as sentences in lines 55 -57, 222 -224.

Author Response

Comment 1. In Figure 1, the enzyme-catalyzed thio-NAD cycling should include more time points. The authors just showed the initial time points for the reaction. Longer time period to show the maximum signal should be included.

Reply 1. If we measure the signal longer, it will saturate. That is, we cannot describe that “a longer period shows the maximum signal”.

Comment 2. This manuscript tried to develop a new tool for NS1 protein detection. Thus, the original detection results to show the signal density should be included in Figures 2 and 3.  

Reply 2. We added the raw data in Supplementary Materials.

Comment 3. Line 58, “testing against IgM and IgG antibodies” was mentioned twice. But the antigen test was missed.

Reply 3. Thank you for pointing out the problem. We revised it.

Comment 4. Lines 61 – 63, “The rapid test …. are relatively low” needs references.

Reply 4. Thank you for pointing out the problem. We added the references.

Comment 5. Line 121, 3a-HSD needs the full name.

Reply 5. We have already mentioned it first in Section 2.2 Reagents and Chemicals. So, we do not need to spell out in this line.

Comment 6. In the discussion part, more new technologies should be discussed. Recently, tandem toehold-mediated displacement reactions (tTMDR) were developed to detect four serotypes of dengue virus with a limit of as low as six copies of RNA per sample (Chem. Commun. 2018, 54, 968–971; ACS Infect. Dis. 2019, 5, 1907–1914).

Reply 6. Thank you for your suggestion, we cited this paper in Discussion and added some explanations. See page 7.

Comment 7. Reference 29 was missed.

Reply 7. Thank you for pointing out the problem. We revised it.

Comments on the Quality of English Language

The manuscript was well written. Some sentences need rewriting to improve the quality of the English language, such as sentences in lines 55 -57, 222 -224.

Reply. We had already asked an American English editor to improve the manuscript.

Thank you for useful comments.

Round 2

Reviewer 1 Report

The changes made by the authors differ somewhat from the reviewer's suggestions. Nevertheless, the new version of the article gives a fairly complete description of the study. Difficulties in improving the article according to some comments can be qualified as objective circumstances beyond the control of the authors. In view of the foregoing, the article can be recommended for publication.

Reviewer 2 Report

I recommend accepting it in its current form.